# Comparative Analysis of Autophagy and Apoptosis in Disc Degeneration: Understanding the Dynamics of Temporary-Compression-Induced Early Autophagy and Sustained-Compression-Triggered Apoptosis

**DOI:** 10.3390/ijms25042352

**Published:** 2024-02-16

**Authors:** Md Abdul Khaleque, Jae-Hoon Kim, Hwan-Hee Lee, Ga-Hyun Kim, Whang-Yong You, Woo-Jin Lee, Young-Yul Kim

**Affiliations:** Department of Orthopedic Surgery, Daejeon St. Mary’s Hospital, College of Medicine, The Catholic University of Korea, Daejeon 34943, Republic of Korea; abdulkhaleque.dream@gmail.com (M.A.K.); superbdoc@daum.net (J.-H.K.); hwanhee337@naver.com (H.-H.L.); rlarkgus21@gmail.com (G.-H.K.); yhy30519@cmcnu.or.kr (W.-Y.Y.); ortho0119@cmcnu.or.kr (W.-J.L.)

**Keywords:** IVDD, NP, autophagy, apoptosis, SC, TC

## Abstract

The purpose of this study was to investigate the initiation of autophagy activation and apoptosis in nucleus pulposus cells under temporary compression (TC) and sustained compression (SC) to identify ideal research approaches in intervertebral disc degeneration. Various techniques were used: radiography (X-ray), magnetic resonance imaging (MRI), transmission electron microscope (TEM), H&E staining, Masson’s trichrome staining, immunohistochemistry (IHC) (LC3, beclin-1, and cleaved caspase-3), and real-time polymerase chain reaction (RT-qPCR) for autophagy-related (beclin-1, LC3, and P62) and apoptosis-related (caspase-3 and PARP) gene expression analysis. X-ray and MRI revealed varying degrees of disc degeneration, ranging from moderate to severe in both groups. The severity was directly linked to compression duration, with SC resulting in notably severe central NP cell degeneration. Surprisingly, TC also caused similar, though less severe, degeneration. Elevated expression of LC3 and beclin-1 was identified after 6 weeks, but it notably declined after 12 weeks. Central NP cells in both groups exhibited increased expression of cleaved caspase-3 that was positively correlated with the duration of SC. TC showed fewer apoptotic markers compared to SC. LC3, beclin-1, and P62 mRNA expression peaked after 6 weeks and declined after 12 weeks in both groups. Cleaved caspase-3 and PARP expression peaked in SC, positively correlating with longer compression duration, while TC showed lower levels of apoptosis gene expression. Furthermore, TEM results revealed different events of the autophagic degradation process after 2 weeks of compression. TCmay be ideal for studying early triggered autophagy-mediated degeneration, while SC may be ideal for studying late or slower-triggered apoptosis-mediated degeneration.

## 1. Introduction

Intervertebral disc degeneration (IVDD) is the leading cause of degenerative spine diseases such as disc herniation, causing back and neck pain, decreasing the quality of life, and augmenting the socioeconomic burden [1]. The generation of IVDD is a complex process involving numerous factors such as biomechanics, aging, genetic factors, nutrition, inflammation, and so forth [2,3]. Due to the complexity of the spine, as well as the multiple factors that contribute to IVDD pathogenesis, no effective conservative treatment is currently available. Most therapeutic strategies focus on symptomatic treatment or in the advanced stages of the disease. Regarding the factors mentioned above, mechanical stress, as part of the biomechanics factors, may contribute to playing a crucial function in the pathogenesis of IVDD predominantly [4,5]. The intervertebral disc (IVD) is the largest veinless, immune-privileged, low-nutrient organ [5,6,7]. It is composed of complex interconnected tissue: the central, gelatinous NP encapsulated by the collagenous laminar annulus fibrosus (AF) and cartilage endplates, providing load, shock neutralization, and movement of the spine [8,9]. Particularly, central gelatinous NP, compared to peripheral AF cells, relies heavily on blood diffusion at disc margins to acquire nutrients. Several factors, such as mechanical stress, injury, smoking, and aging, can reduce nutrient transport to the disc [10]. Reducing nutrient supply is considered one of the key components to begin disc degeneration [6]. Mechanical compression can be crucial in disc degeneration due to these contributing factors. It has been reported that compression-mediated disc degeneration occurs mainly through autophagy and apoptosis, which are the primary programmed cell death pathways for homeostasis in cell survival in the body [11]. It is well established that NP dysfunction is closely associated with IVDD [12,13]. In NP tissue, NP cells are essential in maintaining the extracellular matrix’s homeostasis [8,14]. However, stimulation by external factors, such as oxidative stress, inflammation, or mechanical loading, leads to increased apoptosis and catabolic or anabolic imbalances that eventually trigger and accelerate IVDD progression. Excessive NP cell apoptosis and ECM degradation caused by oxidative stress, inflammation, and mechanical loading are vital pathogenic mechanisms of disc degeneration [15,16,17,18].

Two principal forms of programmed cell death exist, autophagy and apoptotic cell death, which exhibit distinct morphological features. These processes frequently initiate as a response to various stress stimuli. Compression-induced apoptosis has been well-studied in IVD cells. However, the role of autophagy in IVD cells has yet to be documented when IVD cells are exposed to compression. IVDs comprise the surrounding AF and the central gelatinous NP cells [8,9]. Limiting nutrients, such as amino acids, growth factors, and energy, can induce autophagy in IVD of AF cells, but little is known about the role of starvation-induced autophagy in NP cells [6,19,20]. The degeneration of NP cells is associated closely with the etiology of DDD. Excessive or inappropriate compressive force stimulus applied to intervertebral discs is a contributing factor in causing disc degeneration [16]. Compression directly affected the synthesis of collagen and proteoglycan, two extracellular matrix components in IVD cells [3,14]. However, the association between compression and IVD degeneration remains elusive.

Autophagy is a fundamental intracellular process that enables cells to eliminate damaged and dysfunctional components by breaking them down and recycling their constituents [21,22]. This process plays a crucial role in cell survival, as it helps to maintain cellular metabolism and prevents the accumulation of toxic proteins and organelles, particularly under conditions of nutrient-deprivation-induced stress [23,24]. Autophagy involves autophagy-related genes and proteins [22,23]. Under physiological conditions, basal autophagy serves as a quality control mechanism for cellular renovation and homeostasis with relatively low expression levels of Atg proteins. Under stress conditions, Atg proteins are activated for the formation, maturation, and degradation of the autophagosome that captures damaged organelles, misfolded proteins, and invading microorganisms in induced autophagy, leading to the release and re-utilization of its constituents by the fusion with the lysosomes [24,25,26]. Autophagy has also been reported to be implicated in the progression of disc degeneration. The relationship between autophagy and apoptosis is complex, as they share the same set of cellular regulator proteins and are closely linked. Autophagy could either inhibit or delay the occurrence of apoptosis or promote apoptosis [27,28].

Apoptosis, known as type I PCD, is an essential homeostatic mechanism in multicellular organisms, allowing the elimination of no longer needed or seriously damaged cells by an orderly process of cellular disintegration [2,10,29]. It is characterized by caspase activation, cell shrinkage, nuclear and cytoplasmic condensation, DNA fragmentation, and the formation of apoptosomes [27]. Studies have reported that cell loss resulting from apoptosis continues throughout life and plays a vital role in IVD degenerative progression [26]. In addition to apoptosis and autophagy, IVD cells can undergo various other forms of cell death in response to compression [5]. However, recent research has shed light on the critical role of autophagic processes in the progression of central disc degeneration. It is now recognized that autophagy plays a pivotal function in this degenerative process [11,23,30,31,32]. Nevertheless, since the autophagic process initiates during the early stages of disc degeneration, some previous studies may have been unsuitable for capturing these early events. Consequently, we used TCD-7 and TCD-14 for capturing the early events of autophagy, whereas SC groups were utilized to capture late events of apoptosis as well as comparing suitable methods to study them. Thus, we hypothesized to investigate and identify the early events of autophagy as well as late events of apoptosis for studying autophagy and apoptosis-mediated IVDD.

## 2. Results

### 2.1. Temporary and Sustained Compression Induced Radiologically Different Degrees of Disc Degeneration

Both the TC and SC groups were subjected to compression and the resulting disc degeneration was assessed through X-ray imaging. Our findings revealed that both types of compression led to an increase in disc degeneration and promoted the degeneration process in a duration and time-dependent manner. The degeneration pattern observed in the TC group was similar to that in the SC group, while the rate of degeneration was not equivalent. This discrepancy indicates that the effect of compression on disc degeneration is influenced by the duration of compression and the magnitude of the applied load. To further investigate the impact of compression on disc degeneration, we performed MRI at different time intervals following compression. The disc status was evaluated at 2, 4, 6, and 12 weeks after the compression event, revealing a progressive elevation in disc degeneration. Our results demonstrated that disc degeneration increased gradually, with the extent of degeneration being contingent upon the duration of compression and the applied load. Notably, longer compression duration and loads were associated with more pronounced degrees of disc degeneration which is obvious. These findings contribute valuable insights into the relationship between SC and the process of disc degeneration (Figure 1).

### 2.2. Temporary and Sustained Compression Induced Different Degrees of NP Cell Degradation

We investigated the effect of compression on NP cell degeneration using H&E staining. After applying compression to both TC and SC groups, severe NP cell degeneration was observed compared to the Ctrl group. In SC groups, the degeneration gradually elevated with an increase in compression duration. In contrast, the TC group displayed a similar pattern of NP cell degeneration, but the rate of degeneration differed from that in the SC group. Furthermore, upon compression removal, the degeneration process persisted but slowed down. In some instances, the degeneration continued at the same rate observed in the SC group, contrary to our expectations. These findings suggest that the rate of NP cell degeneration is dependent on compression load and time, similarly for both TC and SC. It is noteworthy that once compression is applied, it may have lasting consequences on NP cells, even if the compression is temporary (Figure 2).

Furthermore, to assess the status of NP cell injury after compression, Masson’s trichrome staining was employed. Both experimental groups exhibited marked degeneration of central NP cells following compression. SC resulted in a gradual and load-dependent increase in NP cell degeneration. Conversely, TC also led to a similar pattern of NP cell degeneration, although the rate of degeneration was not as pronounced as that seen with SC. Notably, once NP cell degeneration commenced following compression, it persisted even after the compression was removed, but the degeneration rate was slowed down. These findings suggest that compression-induced NP cell degeneration is a time- and load-dependent process. It is noticeable that once it receives compression, it may have lasting consequences on NP cell morphology, even if the compression is short-term (Figure 2).

### 2.3. Temporary Compression Induced Pronounced Autophagy-Mediated NP Cell Degradation

To assess autophagic activity, we conducted IHC for beclin-1 and LC3. Elevated beclin-1 expression was observed in all three groups compared to the control groups after 6 and 12 weeks, serving as a primary indicator of autophagic initiation in the presence of compression. The initiation rate of autophagy varied among these groups due to compression and duration variation as expected, but the beclin-1 expression elevated after 6 weeks and downregulated after 12 weeks. The SC group consistently exhibited higher beclin-1 expression than the TC group throughout this study. Notably, the TC group displayed upregulated beclin-1 expression even after compression removal for 7 and 14 days, indicating a lasting impact of compression on autophagy-mediated NP cell degeneration (Figure 3).

The presence of the LC3 protein is a major indication of autophagosome maturation. We observed significantly elevated lateral LC3 expression in all three compressed groups after 6 and 12 weeks compared to the control group. LC3 expression was higher in the SC group compared to the TC group after 6 weeks. However, after 12 weeks, expression in the SC group was not significantly different from the TCD-14 group, indicating that TC has a long-lasting impact on NP cell degeneration, whereas the SC group may utilize another pathway leading to severe NP cell death rather than autophagy. This resulted in a decline in the activity of autophagy-mediated degeneration after 12 weeks. These collective pieces of evidence suggest that LC3 expression starts early, peaks at 6 weeks, and is gradually downregulated. Even after SC, LC3 expression declined more in comparison to TC (Figure 3).

Furthermore, we examined the expression of autophagy-related genes in IVD using RT-qPCR for beclin-1, LC3, and P62. The beclin-1 mRNA expression was higher in (TCD-7, TCD-14, and SC) all groups compared to the control group after both 6 and 12 weeks. The SC group exhibited the highest upregulated expression compared to all groups after 6 weeks, followed by the TCD-7 and TCD-14 groups, which showed lower expression than the SC group, as expected. However, the TCD-14 group showed less expression than the TCD-7 group, contrary to our expectations. Nonetheless, these results indicate that beclin-1 mRNA expression is activated early, immediately after the application of compression, peaks after 6 weeks, and then is gradually downregulated after increasing the duration of compression (Figure 4).

LC3 displayed upregulated mRNA expression in the TCD-7, TCD-14, and SC groups compared to the control group. Notably, in the TCD-7, TCD-14, and SC groups, there was significantly upregulated expression after 6 weeks compared to 12 weeks. These findings suggest that autophagy-related catabolic genes are expressed early after experiencing compression, and their expression gradually reduces over time as pressure increases (Figure 4).

P62 mRNA expression was significantly upregulated in the TCD-7, TCD-14, and SC groups compared to the control group after 6 and 12 weeks. Interestingly, the expression was higher after 12 weeks than 6 weeks in the TCD-7 and TCD-14 groups, while it was lower in the SC group, contrary to our expectations. There may be prompt degradation occurring after 12 weeks of SC. These findings suggest that the autophagy pathway may initiate more strongly at an early stage of compression but is downregulated with increasing compression duration (Figure 4).

To investigate the activation of autophagy in NP cell degeneration, we performed TEM to visualize different stages of autophagic events. As we hypothesized that autophagy activates early after applying compression, we used samples subjected to compression for 2 weeks. EM revealed early initiation of phagophore formation, elongation of phagophore formation, autophagosome formation, and autolysosome images. These TEM findings strongly support our hypothesis (Figure 5).

### 2.4. Sustained Compression Induced Different Degrees of Apoptosis-Mediated NP Cell Degradation

To explore the role of apoptosis in NP cell degeneration, we performed IHC for cleaved caspase-3. Cleaved caspase-3 exhibited the highest elevated expression in the central zone of the NP area in the TCD-7, TCD-14, and SC groups after 6 and 12 weeks compared to the control group. Interestingly, in the TCD-7 group, there was no significant upregulation of expression after 12 weeks compared to 6 weeks. In the TCD-14 group, significantly upregulated expression was observed after 12 weeks compared to 6 weeks, although the upregulation was not as pronounced as in the SC group. On the other hand, the SC group displayed the highest and most significantly upregulated expression, surpassing all other groups. These findings suggest that apoptosis markers are expressed later after experiencing more prolonged SC (Figure 3).

Furthermore, to investigate apoptosis-related gene expression, RT-qPCR was performed for cleaved caspase-3 and PARP. Cleaved caspase-3 exhibited significantly elevated mRNA expression in the TCD-7, TCD-14, and SC groups after 6 and 12 weeks compared to the control group. Moreover, upregulated expression was identified in the TCD-7, TCD-14, and SC groups, which was proportionate to the compression duration for the SC group. However, the TCD-7 and TCD-14 groups also displayed the same pattern of expression, albeit significantly less than the SC group, which was an expected outcome. The SC group showed a more pronounced mRNA expression than the TC group. These results suggest that cleaved caspase-3 is positively correlated with compression and duration, which is a major marker for the identification of apoptosis. Thus, the SC group exhibited the highest level of cleaved caspase-3 (Figure 4).

Additionally, the mRNA expression of PARP was analyzed in the TCD-7, TCD-14, and SC groups after 6 and 12 weeks. The mRNA expression showed a significantly upregulated expression in all compression groups compared to the control group, which was proportional to the compression duration for the SC group. The TCD-7 and TCD-14 groups also exhibited the same pattern of expression but significantly less than the SC group. These findings suggest a positive correlation between PARP and compression duration, with the sustained-compression group exhibiting the highest level of PARP. PARP stands for poly adenosine diphosphate-ribose polymerase, a type of enzyme that helps repair DNA damage. So, more DNA fragmentation also proves more PARP which is positively correlated with apoptosis because DNA fragmentation is a major characteristic of apoptosis [33], We found the PARP expression gradually increased with the duration of compression which was highest after 12 weeks in the SC group as expected (Figure 4).

## 3. Discussion

Several factors play a role in IVDD, such as mechanical compression, age, nutrition, and so forth [14]. From them, mechanical compression is considered one of the key contributors to IVDD [34]. Regarding this consideration, we investigated the effects of both TC and SC on disc degeneration, as revealed through X-ray imaging, providing valuable insights into this dynamic pathological process. Our findings expressed that both forms of compression contribute to increased disc degeneration, with the progression dependent on the duration and magnitude of the applied load. This aligns with previous studies demonstrating the impact of compression on disc degeneration [9,35]. To further explore the dynamics of compression-induced disc degeneration, we conducted longitudinal MRI assessments at intervals of 2, 4, 6, and 12 weeks post-compression. The results revealed a progressive escalation in degeneration over time, underscoring the time-dependent nature of compression-induced disc degeneration. Our study established a direct correlation between longer compression duration, higher applied loads, and more pronounced degrees of disc degeneration. These resultssupport the hypothesis that the intensity and duration of compression play pivotal roles in shaping the extent of disc degeneration, consistent with previous research emphasizing the critical contribution of mechanical factors to IVDD [36,37].

In a rodent tail model, static compression was observed to induce cell death in the inner AF, cartilage endplates, and NP, with the process worsening with increased magnitude and duration of loading [9,38]. The results of H&E and Masson’s trichrome staining confirmed that the degeneration of NP cells induced by compression is a time- and load-dependent process. This phenomenon is observed in both TC and SC groups. These findings are consistent with previous studies that have demonstrated the sensitivity of NP cells to mechanical compression, suggesting that prolonged compression can result in cell death and degeneration [39]. Notably, our results suggest that the effects of compression on disc degeneration extend beyond the compression period, persisting even after the removal of compression. This observation is consistent with prior research highlighting the continued degeneration of discs post-loading cessation [39].

LC3, beclin-1, and P62 serve as crucial protein markers for identifying autophagy activation, being present when the autophagy pathway is initiated. Specifically, beclin-1 manifests during the early stages of the autophagy pathway, LC3 during autophagolysosome formation, and P62 as the material designated for degradation in this pathway before undergoing breakdown [40]. Beclin-1, the mammalian counterpart of yeast Atg6, plays a vital role in autophagy, a process of programmed cell survival heightened during periods of cell stress and dampened during the cell cycle [41,42]. Various studies have highlighted that compression induces morphological changes and triggers the expression of biochemical markers associated with autophagy in rat NP cells, leading to autophagosome formation, as evidenced by LC3B expression [11,23]. We identified laterally higher beclin-1 protein expression in the TC and SC groups after 6 weeks than after 12 weeks from IHC staining; additionally, the same pattern of beclin-1 mRNA expression was found from RT-qPCR, suggesting that limited compression induced more beclin-1 protein expression than prolonged compression. This evidence indicates that TCD-7W-6, TCD-14W6, and SCW-6 may induce more beclin-1 expression, serving as an indication of autophagy initiation. However, in TCD-14, the mRNA expression was found to be somewhat different but as expected. However, this may occur because this study was conducted in -vivo, so it may quickly convert to the next steps of autophagy activation that are usual [23,43]. These findings are consistent with previous studies that have reported the activation of autophagy-related genes, including beclin-1, in response to compression in IVDs [41,44]

In addition, the upregulated expression of the LC3 protein was found from IHC after 6 weeks rather than 12 weeks, and the same pattern of autophagy-related gene (mRNA) was observed in all compression groups, indicating that autophagy is activated early in response to compression, peaked after 6 weeks, and then gradually downregulated as pressure increased. This result was consistent with the study by Zhao et al. 2019, which reported an increase in LC3-II expression in the early stages of compression in IVDs [45].

The upregulated expression of the P62 protein and the autophagy-related gene was also observed in all compression groups, but the expression level was higher after 12 weeks compared to 6 weeks in the TCD-7 and TCD-14 groups, while it was lower in the SC group, which was a little dissimilar to the expected outcome. This result may come because it may quickly be destroyed with autolysosomes. However, this was an in-vivo study, so it would not completely mimic the in- vitro conditions, though the expression pattern was the same, showing an increase after 12 weeks rather than 6 weeks. This result was consistent with a previous study that mentioned that an in-vitro condition, autophagy induction resulted in LC3-II increases and p62/SQSTM1 decreases under serum deprivation [10,43]. However, after prolonged compression, the p62/SQSTM1 expression decreased, indicating that this adaptor protein during autophagosome degradation may reflect another pathway activated to suppress the autophagy induced by unphysiological mechanical loading [40]. These findings suggest that the autophagy pathway may be initiated more strongly in the early stages of compression and downregulated with increasing compression duration.

Furthermore, TEM results unequivocally validated that TC promptly induced autophagy, even immediately after the application of compression. This was evident from the identification of distinct events of autophagic degradation processes observed after two weeks of compression period. Specifically, we visualized the initiation of phagophore formation, elongation of phagophores, autophagosome formation, and the presence of autolysosomes. These findings serve as compelling evidence, affirming that temporary or short-term compression indeed triggers autophagy, a conclusion consistent with prior research in the field. This observation aligns with the work of previous researchers who reported that compression is associated with the initiation of autophagic processes [11]. The confirmation of autophagy induction in the early stages of compression contributes to our understanding of the cellular responses to mechanical stimuli, shedding light on the intricate relationship between compression and autophagic pathways.

The findings of this study also highlighted the importance of compression duration in inducing apoptosis. A study by Chen et al. (2019) investigated the effects of dynamic compression on IVD cells and found that longer compression duration resulted in greater levels of apoptosis [46]. This result is consistent with the results of this study, which found that SC expressed a higher level of cleaved caspase-3, a key apoptotic marker protein in IHC staining. It was gradually increased with increasing duration of compression, whereas TC also increased the rate of cleaved caspase-3 expression but not significantly. We found an upregulated apoptosis-related gene of (mRNA) caspase-3 expression, which was upregulated with increasing compression duration and more expressed after 12 weeks than after 6 weeks. Moreover, the upregulating PARP (mRNA) expression was identified, which was highest in SC groups after 12 weeks than 6 weeks. This evidence also proved that SC could induce apoptosis in a dose- and time-dependent manner. Overall, these findings suggest that short-term compression is a significant contributor to inducing autophagy, whereas long-term compression induces apoptosis in IVD cells. However, a rat model may not precisely replicate human disc degeneration processes. Autophagy inhibitors had not been used to understand the response of compression in -vivo conditions; this would be assessed in the future. Nonetheless, this finding has a potential impact on addressing spine disorders.

## 4. Materials and Methods

### 4.1. Animal Preparation

A total of 34 male Sprague Dawley rats (KOATECH Korea Animal Technology, Pyungtaek, Republic of Korea) aged 9 weeks and weighing between 310 and 360 g were included in this study. The experimental protocol was approved by the Institutional Review Board of the Animal Experimentation Committee (IRB number CMCDJ-AP-2022-002). The rats were randomly allocated into three groups: the control (Ctrl) group, the temporary-compression (TC) group, and the sustained-compression (SC) group. The Ctrl group comprised six rats, while the TC and SC groups consisted of eighteen and eight rats, respectively. For the TC and SC groups, an external fixator was utilized. These groups were further divided into subgroups based on the duration of compression: 6 weeks and 12 weeks (Figure 6B). Following intraperitoneal anesthesia, two cross-shaped 0.7 mm diameter Kirschner wires were percutaneously inserted into each vertebral body perpendicular to the axis of the tail. These wires were attached to aluminum rings (Figure 6A). The rings were longitudinally connected using three threaded rods. The surgical procedures were performed by a registered orthopedic surgeon in a controlled environment. Post-surgery, the rats were housed in a 12 h light/dark cycle with a controlled temperature of 23 ± 2 °C and humidity of 55 ± 5%. They were individually housed in pathogen-free housing units and provided free access to food and water. Two different loading designs were employed: temporary compression day-7 (TCD-7) involved a 7-day compression followed by unloading for up to six weeks and up to twelve weeks, and temporary compression day-14 (TCD-14) included a 14-day compression followed by unloading for up to six weeks and up to twelve weeks (Figure 6B).

### 4.2. Radiography (X-ray)

To assess disc degeneration, X-ray radiography was conducted at regular intervals of 1, 2, 4, 6, and 12 weeks following compression. Prior to the procedure, meticulous attention was given to the administration of anesthesia, ensuring precise dosage based on the rats’ individual body weights to induce muscle relaxation in the caudal region. After successful anesthesia induction, the rats were positioned in a prone posture on a radiographic imaging unit equipped with a molybdenum target (ZS-1001, Shimadzu Corp., Kyoto, Japan). Their tails were positioned in a straight manner on the platform, and radiographs were captured according to the prescribed protocol.

### 4.3. MRI

MRI was conducted at intervals of 2, 4, 6, and 12 weeks following the application of compression. Rats were subsequently sacrificed, and the entire tail was excised for MRI examination. Images were acquired using a 3.0 T MRI machine (Philips, Achieva, Houten, the Netherlands), employing a dedicated coil designed for small animals. The tail was positioned within a tube containing a 0.1 M CuSO4 solution to mitigate susceptibility effects and enhance image contrast. A 2-D spin-echo dual-echo sequence was utilized with the following parameters: repetition time = 9000 ms, echo times = 16 and 80 ms, flip angle = 90, number of averages = 2, slice thickness = 0.6 mm, field of view = 40,640 mm, and in-plane resolution = 0.1 mm. Thirty sagittal slices were obtained. Signal intensity calculations were performed using the T2-weighted image (echo time = 80 ms) for improved visualization, serving as an indirect measure of disc hydration. It is well established that a reduction in water content is a common manifestation of IVDD. The mean signal intensity (brightness) in the control disc was established as the reference for evaluating the signal intensity of the injured discs in each tail (Figure 1).

### 4.4. Transmission Electron Microscope

The experimental segment of the rat’s tail was meticulously excised and immediately placed in a 4% paraformaldehyde solution, where it was allowed to remain at room temperature for duration of 48 days. Subsequently, the excised tail was rinsed with water and subjected to immersion in decalcification reagents, specifically ethylenediaminetetraacetic acid (EDTA), until complete decalcification was achieved. Samples were fixed with 2% glutaraldehyde–2% paraformaldehyde in 0.1 M phosphate buffer (pH 7.4) for 24 h at 4 °C and washed in 0.1 M phosphate buffer, post-fixed with 1% OsO_4_ in 0.1 M phosphate buffer for 1.5 h. The samples were dehydrated in increasing concentrations of ethanol and then infiltrated and embedded with a Poly/Bed 812 kit (Polysciences, Warrington, PA, USA) and polymerized at 60 °C for 18 h. The specimens were sectioned (200 nm) with a diamond knife in the Leica Ultracut microtome (EM-UCT, Leica, Bellevue, WA, USA) and stained toluidine blue for observation of optical microscope. Ultrathin sections (70 nm) were cut using a Leica Ultracut microtome (EM-UCT, Leica, USA) and then double stained with uranyl acetetate and lead citrate in a Leica EM Stainer (AC-20, Leica, USA). The stained samples were examined in a JEM-1011 transmission electron microscope (Jeol, Tokyo, Japan) using an accelerating voltage of 80 kV. Digital images were obtained using the 2 K digital CCD camera (RADIUS, EMSIS, Muenster, Germany).

### 4.5. Paraffin-Embedded Tissue Preparation

The experimental segment of the rat’s tail was meticulously excised and immediately placed in a 4% paraformaldehyde solution, where it was allowed to remain at room temperature for duration of 48 h. Subsequently, the excised tail was rinsed with water and subjected to immersion in decalcification reagents, specifically ethylenediaminetetraacetic acid (EDTA), for 30 days. The decalcified tail was then vertically sectioned to an appropriate size for embedding in a plastic block and subsequently immersed in a 10% formaldehyde solution overnight. The subsequent day, a series of dehydration steps were undertaken to remove water content from the sample. Each section was carefully sliced to a thickness of approximately 5 μm using a microtome and mounted onto slides, following a standardized protocol. Finally, the prepared slides were left to dry overnight and stored at room temperature until they were ready for use in further analyses.

### 4.6. H&E Staining

The paraffin-embedded sections were subjected to deparaffinization using a standardized protocol involving specific deparaffinization steps. Subsequently, hematoxylin and eosin staining, a commonly employed technique in histopathology, was conducted following a standard procedure. The stained sections were then visualized and captured using an OLYMPUS BX53 U-CMAD3 microscope (T7 Tokyo, Japan).

### 4.7. Masson’s Trichrome Staining

Formalin-fixed paraffin-embedded sections underwent a series of dehydration steps to remove water content. Subsequently, the slides were immersed in Bouin’s solution and allowed to incubate at room temperature overnight. On the following day, the slides were thoroughly washed with tap water until the yellow coloration disappeared. The slides were then subjected to a staining procedure, beginning with Weigert’s hematoxylin for duration of 15 min. After rinsing with water, the slides were immersed in Biebrich scarlet-acid fuchsin solution for a period of 5–10 min. Following another water rinse, the slides were placed in a phosphomolybdic–phosphotungstic acid solution contained within a plastic Coplin jar for 15 min. Subsequently, the slides were rinsed with water and stained with aniline blue solution for 10 min. Following a final water rinse, the slides were carefully mounted using appropriate cover slips, taking into consideration dehydration steps. The prepared slides were then observed and documented using an OLYMPUS BX53 U-CMAD3 microscope (T7, Tokyo, Japan).

### 4.8. Immunohistochemistry (IHC)

The sections were subjected to deparaffinization using xylene, followed by dehydration in a series of graded ethanol solutions. Distilled water was used for a 5 min wash. Antigen retrieval was performed by treating the sections with proteinase K enzyme. Following a rinse with distilled water, blocking solution was applied to the sections. Incubation with primary antibodies was carried out overnight at 4 °C. The primary antibodies used included a rabbit monoclonal antibody LC3 (dilution: 1:100; Novus Biologicals, Centennial, CO, USA), a mouse monoclonal antibody beclin-1 (dilution: 1:1000; Abcam, Waltham, MA, USA). Additionally, a mouse monoclonal antibody cleaved caspase-3 (dilution: 1:400; Cell Signaling Technology, Danvers, MA, USA) was utilized and incubated overnight at 4 °C. On the subsequent day, the slides were treated with peroxidase-labeled anti-mouse or anti-rabbit antibodies (VECTASTIN Elite ABC Kit, Vector Laboratories, Inc., Newark, CA, USA) at room temperature for 30 min. HRP detection solutions (ImmPACTNovaRED Substrate kit peroxidase, Vector Laboratories, Inc., Newark, CA, USA) were added and incubated for 15 min. Counterstaining was performed using Mayer’s hematoxylin, followed by dehydration steps. Finally, appropriate cover slips were used for mounting the slides, and imaging was conducted using an OLYMPUS BX53 U-CMAD3 microscope (T7, Tokyo, Japan).

### 4.9. Quantitative Real-Time Polymerase Chain Reaction (RT-qPCR)

The experimental intervertebral discs, including the AF) and NP, were carefully excised and immediately preserved in liquid nitrogen to prevent RNA degradation. Subsequently, the specimens were stored at −80 °C until further analysis. Total RNA extraction was performed using the RNeasy Mini Kit (Qiagen, Hilden, Germany). Subsequently, 1 μg of RNA was reverse transcribed using the Revertra Ace PCR RT Master Mix Kit (TOYOBO, Osaka, Japan). To assess the relative mRNA expression levels of autophagy-related genes (LC3, beclin-1, and P62) and apoptosis-related genes (caspase-3 and PARP), real-time polymerase chain reaction (PCR) was conducted. The Applied Biosystems 7500 Fast Real-Time PCR System (Thermo Fisher Scientific, Waltham, MA, USA) and Thunderbird SYBR Green PCR Mix (TOYOBO, Osaka, Japan) were employed for the PCR analysis. Primer sequences used for amplification are provided in Table 1. The mRNA expression of the target genes was normalized to the endogenous control, actin mRNA expression. The relative mRNA expression levels of each protein in the experimental discs were determined using the 2^−ΔΔCt^ method. The difference in threshold cycles (ΔCt) between the target gene and the reference gene (GAPDH) was calculated using the formula ΔCt = Cttarget gene—CtGAPDH. The mRNA expression fold change in the target gene, represented as 2^−ΔΔCt^, was calculated to indicate the relative amount of mRNA compared to the control group.

### 4.10. Statistical Analysis

Data were expressed as mean ± standard deviation, and two-way ANOVA was used to investigate changes the effects of TCD-7, TCD-14, and SC groups after six and twelve weeks. Six rats were used for the Ctrl group, fourteen rats for TCD-7 and TCD-14 groups, as well as fourteen rats for SC group. Statistical significance was assessed with significance levels of *p* < 0.05, *p* < 0.01, and *p* < 0.001 (significance levels are indicated by * *p* < 0.05, ** *p* < 0.01, and *** *p* < 0.001) using GraphPad Prism 9.5.1 (GraphPad Software, Inc., La Jolla, CA, USA).

## 5. Conclusions

Temporary compression or short-term compression triggers early autophagy-mediated degeneration, whereas sustained compression or long-lasting compression triggers slower apoptosis-mediated degeneration. So, it can be concluded that temporary compression may suit autophagy study and sustained compression may suit apoptosis study in IVDD.

## Figures and Tables

**Figure 1 ijms-25-02352-f001:**
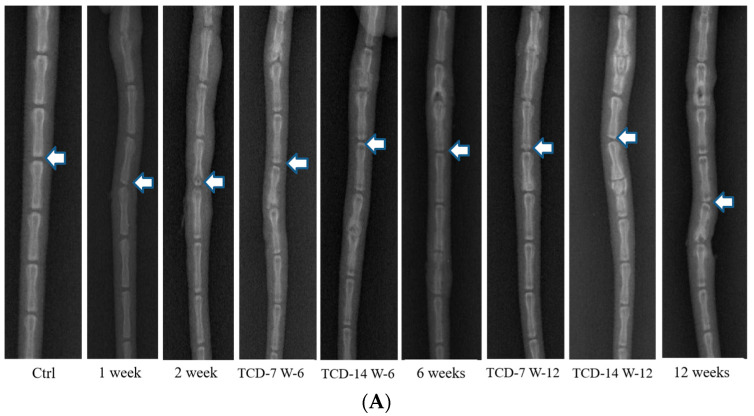
(**A**) In X-ray imaging, compression led to an increase in disc degeneration, with a time-dependent and load-specific influence. The degeneration pattern in both temporary- and sustained-compression groups was similar. However, there was a notable difference in the rate of degeneration (White arrow: Indication of disc status after compression and control). (**B**) In MRI, observations at different intervals (2, 4, 6, and 12 weeks) post-compression revealed a progressive elevation in disc degeneration over time (White arrow: Indication of disc status after compression and control).

**Figure 2 ijms-25-02352-f002:**
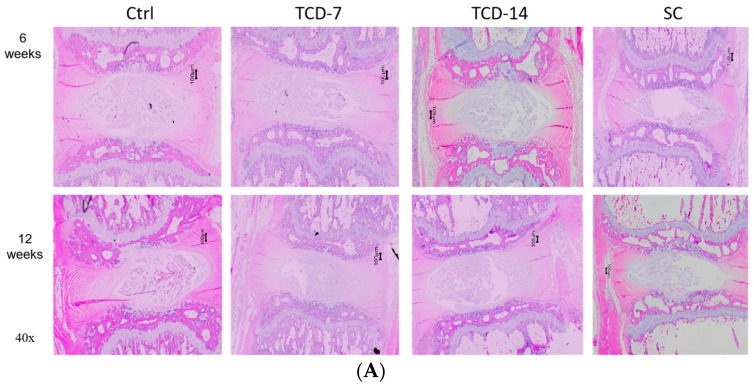
(**A**) H&E staining and (**B**) Masson’s trichrome staining: The severity of degeneration was directly related to the duration of compression, with SC leading to notably severe central NP cell degeneration, which increased with the duration of compression. Unexpectedly, TC also led to similar but less severe degeneration.

**Figure 3 ijms-25-02352-f003:**
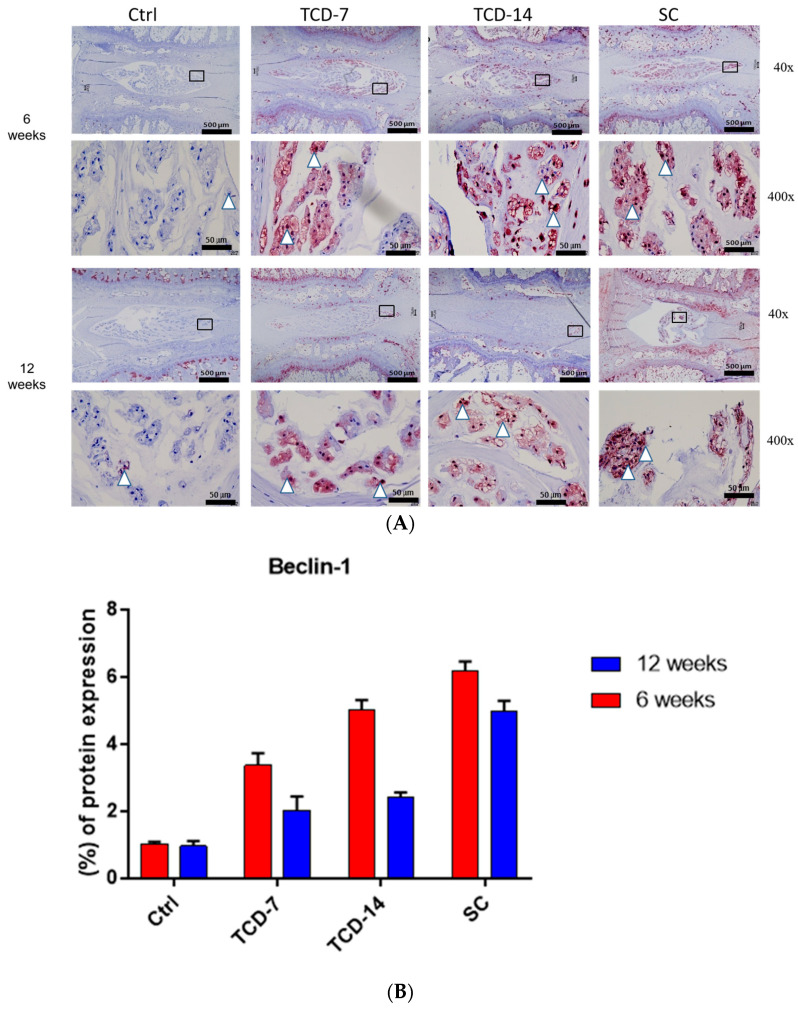
(**A**,**B**) Beclin-1 expression elevated after both 6 and 12 weeks in all groups compared to the control group. However, the expression was more elevated after 6 weeks than after 12 weeks in the TCD-7, TCD-14, and SC groups. (**C**,**D**) LC3 expression was elevated after both 6 and 12 weeks in all groups compared to the control group. However, the expression was more elevated after 6 weeks than after 12 weeks in the TCD-7, TCD-14, and SC groups. (**E**,**F**) Cleaved caspase-3 expression was elevated in all groups after 6 and 12 weeks compared to the control group. However, the expression was not significant in the TCD-7 and TCD-14 groups after 12 weeks than after 6 weeks. On the contrary, the expression was peaked and more significant after 12 weeks than after 6 weeks in the SC group (black squares: area is to be magnified, white triangles: indication of the immunopositive area).

**Figure 4 ijms-25-02352-f004:**
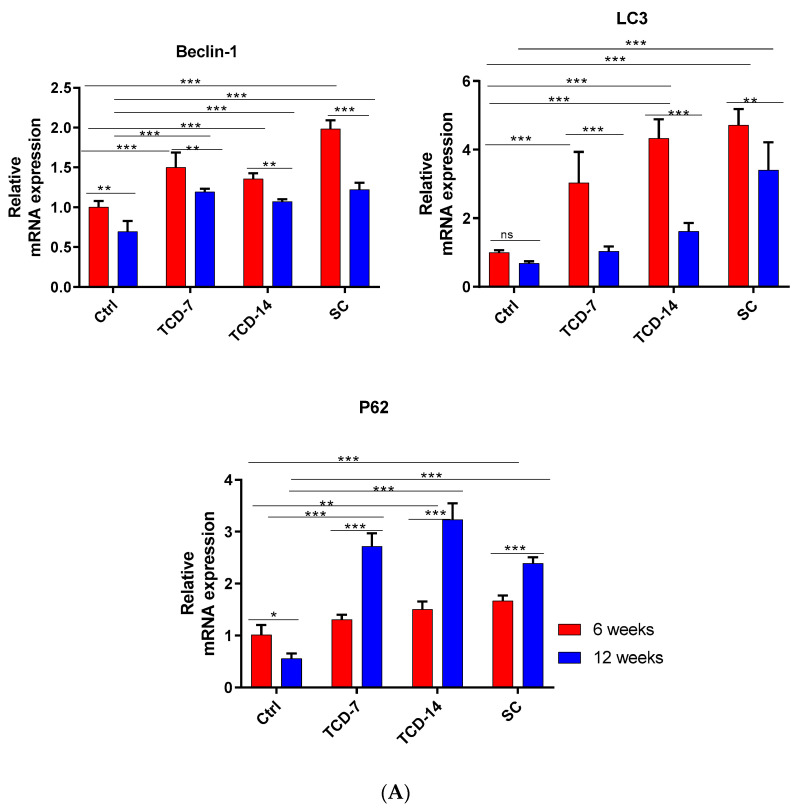
(**A**) Autophagy-related gene mRNA expression (RT-qPCR): The mRNA expression of beclin-1 was found to be elevated in all groups after both 6 and 12 weeks compared to the control group. However, after 6 weeks, the expression was more elevated than after 12 weeks. LC3 mRNA expression showed an increase after 6 weeks in all groups compared to the control group. However, after 12 weeks, the expression was downregulated compared to the levels observed at 6 weeks and was somewhat lower than the control group. P62 expression was elevated after both 6 and 12 weeks in all groups compared to the control group. Interestingly, after 12 weeks, the expression was more upregulated than at 6 weeks. In the SC group, contrary to expectations, the expression was more downregulated after 12 weeks compared to the TCD groups (level of significance is * *p* < 0.05, ** *p* < 0.01, and *** *p* < 0.001). (**B**) Apoptosis-related gene mRNA expression (RT-qPCR): The mRNA expression of PARP was elevated in all groups after both 6 and 12 weeks compared to the control group. Notably, after 12 weeks, a more elevated expression was observed than after 6 weeks. In the TCD-7 and TCD-14 groups, there was no significant upregulation, whereas the SC group exhibited statistically significant upregulation after 12 weeks compared to 6 weeks. Caspase-3 mRNA expressions showed an elevation after both 6 and 12 weeks in all groups compared to the control group. However, after 12 weeks, a more upregulated expression was identified than at 6 weeks. The SC group exhibited the highest expression compared to the TCD-7 and TCD-14 groups (level of significance is * *p* < 0.05, ** *p* < 0.01, and *** *p* < 0.001) (ns: No significance).

**Figure 5 ijms-25-02352-f005:**
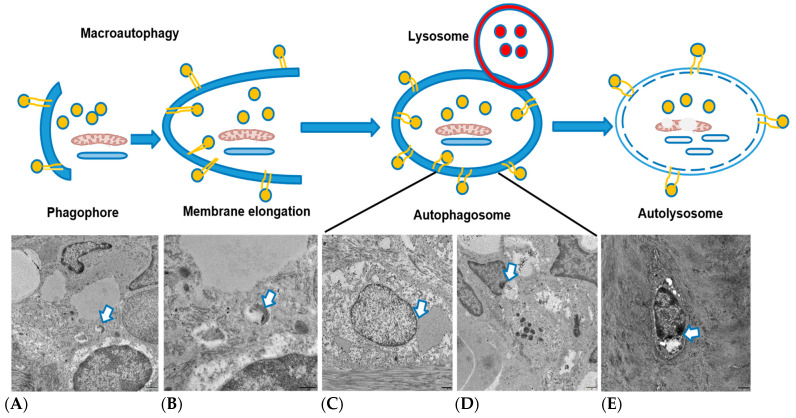
Visualization of different events of autophagy activation after 2 weeks of compression: (**A**) phagophore formation, (**B**) membrane elongation, (**C**) autophagosome, (**D**) autolysosome, and (**E**) degradation (White arrow: Indication of different events of autophagy).

**Figure 6 ijms-25-02352-f006:**
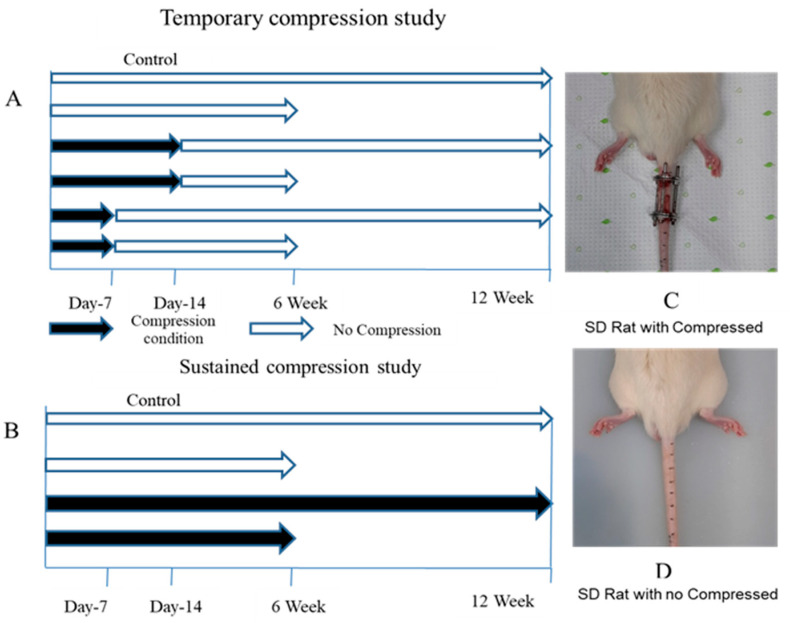
Compression plan as well as post-operative and pre-operative stages of rats: (**A**) temporary-compression plan, (**B**) sustained-compression plan, (**C**) SD rat with compression, and (**D**) SD rat without compression considered as control.

**Table 1 ijms-25-02352-t001:** List of primers.

Primer Name	Primer Sequence
LC3	Forward 5′-CCTGCTGCTGGCCGTAGT-3′ Reverse 5′-TGATGAAGTCTTCCTGCCAAAA-3′
Beclin-1	Forward 5′-TTCAAGATCCTGGACCGAGTGAC-3′ Reverse 5′-AGACACCATCCTGGCGAGTTTC-3′
P62	Forward 5′-TCCCTGTCAAGCAGTATCC-3′ Reverse 5′-TCCTCCTTGGCTTTGTCTC-3′
Caspase-3	Forward 5′-GTGGAACTGACGATGATATGGC-3′ Reverse 5′-CGCAAAGTGACTGGATGAACC-3′
PARP	Forward 5′-CGGCACGAGAGGGAGGATGG-3′, Reverse 5′-TGTCAGGCTGCCGGATGGAGT-3′
GAPDH	Forward 5′-GTATCGGACGCCTGGTTAC-3′ Reverse 5′-CTTGCCGTGGGTAGAGTCAT-3′

## Data Availability

Datasets are available through the corresponding author upon reasonable request.

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
