# Peer review of "Comparative Analysis of Autophagy and Apoptosis in Disc Degeneration: Understanding the Dynamics of Temporary-Compression-Induced Early Autophagy and Sustained-Compression-Triggered Apoptosis"

_ijms, 2024, doi:10.3390/ijms25042352_

Round 1

Reviewer 1 Report

Comments and Suggestions for Authors

In this manuscript, the authors examined the molecular regulatory mechanisms underlying compression-mediated disc degeneration. The authors claimed that autophagy and apoptosis are induced in cartilage by compression using immunohistochemistry and realtime RT-PCR. This manuscript is of interest from the point of elucidating the molecular regulatory mechanisms underlying compression-mediated disc degeneration, though there are some concerns, and they are discussed below.

1. no gain of function or loss of function experiments.

              The authors examined the phenomenon after loading with immunohistochemistry and realtime RT-PCR, but no intervention such as the use of autophagy inhibitors. The reviewer is curious what will happen in case autophagy inhibitor was applied during the application of compression.

2. The quality of the stained histological section can be improved, especially the immunohistochemical staining.

3. As for immunohistochemical staining, negative staining, no primary antibody but use of secondary antibody, is mandatory to evaluate nonspecific binding of the secondary antibody.

4. p-6, l-3, “a 4% paraformaldehyde solution, where it was allowed to remain at room temperature for duration of 48 days.” might be 48 hours, not days.

5. p-6, l-5, “specifically ethylenediaminetet-raacetic acid (EDTA), until complete decalcification was achieved.” please describe detailed duration such as “about 30 days”.

Author Response

I have attached the response herewith. Please find the attached file.

Reviewer 2 Report

Comments and Suggestions for Authors

Authors present an animal rat study on 34 rats , who were allocated randomly to 3 groups: Control (Ctrl) group, the Temporary Compression (TC) group, and the Sustained Compression (SC) group, with two different loading desings: emporary Compression Day-7 (TCD-7) involved a 7-day compression followed by unloading for up to six weeks and up to twelve weeks. Temporary Compression Day-14 (TCD-14) in order to investigate the initiation of autophagy activation and apoptosis in nucleus pulposus cells under TC and SC and to identify ideal research approaches in intervertibral disc degeneration. The severity of disc degeneration in X-ray and MRI correlated with compression duration; LC3, Beclin-1, P62 mRNA expression were peaked after 6 weeks and declined after 12 weeks in both groups. Cleaved caspase-3, and PARP expression were peaked in SC, positively correlating with longer compression duration, while TC showed lower levels of apoptosis gene expression. Authors conclude that  TC may ideal for studying early triggered autophagy-mediated degeneration, while SC may ideal to study late or slower triggered apoptosis-mediated degeneration.

Authors should clearly define neuroradiological parameters of degeneration in the X-ray and MRI-imaging, i.e. how was it assessed und who did the assessment. Clear limitation is that this is an animal study and that the results are not easily reproducible and transmissible to humans; however, what this study lacks are future advances and potential clinical applications of these findings. Role of autophagy and investigated parameters should be more thoroughly explained in the literature review. 

Comments on the Quality of English Language

Acceptable. 

Author Response

I have attached the response herewith. Please find the attachment.

Round 2

Reviewer 2 Report

Comments and Suggestions for Authors

Authors have sufficiently responded to reviewers remarks. 

Comments on the Quality of English Language

Acceptable.